# TRIM32 and Malin in Neurological and Neuromuscular Rare Diseases

**DOI:** 10.3390/cells10040820

**Published:** 2021-04-06

**Authors:** Lorena Kumarasinghe, Lu Xiong, Maria Adelaida Garcia-Gimeno, Elisa Lazzari, Pascual Sanz, Germana Meroni

**Affiliations:** 1Instituto de Biomedicina de Valencia, CSIC, Jaime Roig 11, 46010 Valencia, Spain; lkumarasinghe@ibv.csic.es; 2Department of Life Sciences, University of Trieste, Building Q, Via L. Giorgieri 5, 34127 Trieste, Italy; lu.xiong@phd.units.it (L.X.); elazzari@units.it (E.L.); 3Department of Biotechnology, Polytechnic University of Valencia, 46022 Valencia, Spain; magar27m@btc.upv.es; 4Centro de Investigación Biomedica en Red Enfermedades Raras CIBERER-ISCIII, 46010 Valencia, Spain

**Keywords:** TRIM proteins, E3-ligase, Trim32, malin, ubiquitination, Limb-Girdle Muscular Dystrophy, Lafora disease, rare diseases

## Abstract

Tripartite motif (TRIM) proteins are RING E3 ubiquitin ligases defined by a shared domain structure. Several of them are implicated in rare genetic diseases, and mutations in TRIM32 and TRIM-like malin are associated with Limb-Girdle Muscular Dystrophy R8 and Lafora disease, respectively. These two proteins are evolutionary related, share a common ancestor, and both display NHL repeats at their C-terminus. Here, we revmniew the function of these two related E3 ubiquitin ligases discussing their intrinsic and possible common pathophysiological pathways.

## 1. Introduction

### 1.1. Ubiquitination

The post-translational modification of proteins with the small peptide ubiquitin can regulate target protein turnover, subcellular localization, or activity, depending on the topology of the ubiquitin chain built on the substrate. Given these broad effects, the process occurs through a tightly regulated enzymatic cascade. In the first step, ubiquitin is activated in an ATP-dependent manner by the E1 activating enzyme. Ubiquitin is then transferred to the active site of an E2 ubiquitin-conjugating enzyme, which in turn cooperates with an E3 ubiquitin ligase for the final transfer of ubiquitin onto the target protein [1,2] (Figure 1). While only two types of E1 enzymes are found, there are about 40 E2 enzymes and, in humans, more than 600 E3s [3]. Substrate specificity is guaranteed by the existence of such a large number of E3 ubiquitin ligases.

The addition of one ubiquitin on a residue of the target protein results in monoubiquitination, while multiubiquitination is the process of adding single ubiquitin moieties to different residues of the target protein. Polyubiquitination is the building of a poly-ubiquitin chain on a single residue of the target (Figure 1). In the latter case, the C-terminus of the incoming ubiquitin molecule binds to one of the seven lysine residues (K6, K11, K27, K29, K33, K48 or K63) or the N-terminal methionine (M1) of the previous ubiquitin molecule, thus forming chains of different topologies [4] (Figure 1). The type of chain that is formed, its length, and the type of residue of the substrate targeted with the poly-ubiquitin chain will influence the pathway undertaken by the modified substrate [5]. For instance, it is well-established that proteins with K48-linked poly-ubiquitin chains are directed to proteasomal degradation, while K63-linked poly-ubiquitin chains are involved in the regulation of processes such as DNA repair, protein trafficking, or RNA translation [6,7].

### 1.2. E3 Ubiquitin Ligases

There are three main classes of E3 ligases that differ in terms of domain types and the modality of ubiquitin transfer to the target protein. The largest class of ubiquitin ligase includes the RING E3s. For transferring ubiquitin to the substrate, this class of E3 ligases acts as a scaffold for the ubiquitin-charged E2 enzyme to favor a direct passage of ubiquitin from the E2 to the substrate [8,9,10] (Figure 1). In the case of RING E3 ligases, the E2 conjugating enzyme determines which of the lysine residues on ubiquitin is used to build the poly-ubiquitin chain, allowing for different regulatory outcomes, while the E3 ligase interacts with the target substrate, thus providing specificity to the process. The other types of E3 ligases are represented by HECT (homologous to the E6AP carboxyl terminus) and RBR (RING-between RING-RING) E3s. In HECT ligases, the transfer of ubiquitin from the E2 to the substrate occurs through a two-stage process allowed by the bilobar architecture of these proteins. Ubiquitin is transferred from the E2 to the catalytic cysteine on the E3, contained in the conserved C-terminal lobe, followed by transferring to the substrate bound to the N-terminal lobe. The two lobes are linked by a flexible hinge, allowing them to assume an orientation aimed at favoring the transfer of ubiquitin [10,11] (Figure 1). Finally, the RBR family of E3 is characterized by three domains: RING1 and RING2 domains at the sides and an In-Between RING (IBR) domain at the center. RING1 recruits the ubiquitin-charged E2, while the catalytic cysteine is found within the RING2 domain. The IBR domain is similar to the RING2 domain but lacks the catalytic site. This class of E3s transfers the ubiquitin to the substrate in the same two-step process described for the HECT E3 ligases [10,12] (Figure 1).

### 1.3. TRIpartite Motif (TRIM) Ubiquitin E3 Ligases

The RING-type E3 ligases can work as multiprotein complexes, as observed for RING-Cullin ligases where the isolated catalytic RING domain associates with other subunits to form an active catalytic complex targeting specific substrates, or can act as single proteins composed of different functional domains with specific catalytic and substrate recognition functions [9]. In the latter category, the TRIpartite Motif (TRIM) family is characterized by the presence of an N-terminal three-domain module composed of the catalytic RING domain followed by one or two B-box domains and a coiled-coil region that mediates oligomerization [13]. In most TRIM proteins, the C-terminal domain mediates substrate interaction. As such, this domain is variable and can be represented by COS domain, fibronectin type III repeat (FNIII), PRY domain, SPRY domain, acid-rich region (ACID), filamin-type IG domain (FIL), NHL domain, PHD domain, bromodomain (BROMO), Meprin and TRAF-homology domain (MATH), ADP-ribosylation factor family domain (ARF), or transmembrane region (TM) [14]. The high variability of this domain, which allows the classification of TRIM proteins into 12 different classes, accounts for the large number of substrates targeted by the various members of this family.

By forming specific E2-RING E3-substrate complexes, poly-ubiquitin chains of different topologies can be built on target proteins and differentially affect their activity, stability, or subcellular localization. Indeed, TRIM-mediated ubiquitination regulates a large number of cellular processes, and mutations in TRIM family members often have dramatic effects that result in developmental defects, neurodegeneration, or cancer. In this review, we will focus on two rare genetic diseases: Limb-Girdle Muscular Dystrophy type R8 (LGMDR8, formerly known as type 2H), due to mutations in *TRIM32*, and Lafora disease (LD), caused by mutations in the *EPM2B/NHLRC1* gene encoding the TRIM-like protein malin.

### 1.4. Limb-Girdle Muscular Dystrophy Type R8 and Lafora Disease

LGMDR8 is a rare autosomal recessive genetic disease characterized by the progressive wasting of muscles. Proximal muscles of the arms and legs are the most affected and, although symptoms are highly variable, in severe forms of the disease, patients may require the use of a wheelchair [15,16]. LGMDR8 was originally reported in the Hutterite population of Manitoba, Canada. As an autosomal genetic disease, it affects males and females in equal numbers. The age of onset can vary greatly even among individuals of the same family. The relative frequency of LGMDR8 is unknown, but worldwide, LGMDR8 cases are extremely rare. Histological and ultrastructural analyses of muscular tissue from LGMDR8 patients and *Trim32* knock-out (KO) mice models show similar myopathic phenotypes. Muscular tissue presents disorganized sarcomeres, an increased number of fibers with multiple centrally located nuclei, fiber splitting, abnormal fiber diameter variability, a dilated sarcotubular system with abnormal accumulation of membranous structures, and Z-line streaming [17,18]. Some LGMDR8 patients present neurological features such as neuropathic electromyography components, paresthesia, paresis, and hypoactive tendon reflexes [16,19,20]. In keeping with these observations, *Trim32* KO mice showed a decrease in the concentration of neurofilament proteins in the brain and a reduction in the motor axon diameter [17]. Interestingly, *TRIM32* mutations are identified in both LGMDR8 and sarcotubular myopathy (STM) patients. STM shares similar myopathic phenotypes with LGMDR8, but is distinguished from the latter by its unique muscle structural features, characterized by the presence of abnormal vacuoles arising from the sarcoplasmic reticulum. The common mutations found in LGMDR8 and STM suggest that these two conditions represent different forms of the same disease [16,21]. Taken together, these data would suggest that TRIM32 plays a role in maintaining the homeostasis of both muscles and motor neurons, and its mutations lead to progressive degeneration of the neuromuscular system.

LD is an autosomal recessive rare neurological disorder enclosed in a group of diseases known as progressive myoclonus epilepsies (PMEs) [22]. It has a prevalence of fewer than 4 patients in 1,000,000 individuals. LD occurs most frequently in Mediterranean countries such as Spain, France, and Italy, and other parts of the world such as North Africa, the Middle East, and India’s southern regions. About 50–60% of patients with LD carry mutations in the *EPM2A* gene encoding the glucan phosphatase laforin, while 40–50% carry mutations in the *EPM2B/NHLRC1* gene encoding the E3-ubiquitin ligase malin [23,24]. Malin is detected in complex with laforin, and alterations that occur at the level of one of the two proteins are sufficient to alter the functional harmony of the complex [25]. In support of this, patients with mutations occurring along *EPM2A* or *EPM2B* genes exhibit similar pathological phenotypes despite the different roles the individual proteins play [26]. LD patients suffer from myoclonic episodes and seizures (epilepsy) and present progressive neurodegeneration. The early symptoms of the disorder, such as behavioral changes, depression, and dysarthria, appear in late childhood or adolescence, and the affected worsen in condition with time, with their survival rate estimated to be 10 years from symptom onset. The disease is characterized by abnormal glycogen aggregates, called Lafora bodies (Figure 2). In 1911, the Spanish neurologist Dr. Gonzalo Rodriguez Lafora was the first to report the presence of intense dark inclusions in postmortem preparations of affected patients; he first described these inclusions as amyloid bodies [27]. Although Lafora bodies are a hallmark of the disease, genetic testing is necessary to diagnose the disease after the first clinical manifestations [28].

## 2. Structural Data and Biochemical Function of TRIM32 and Malin

### 2.1. Domain Composition

Both malin and TRIM32 are single-subunit ubiquitin ligases characterized by an N-terminal RING domain, which confers catalytic activity, and a C-terminal domain represented in both proteins by 6 NHL (present in NCL1, HT2A, and LIN-41 proteins) repeats.

In the TRIM32 protein, composed of 653 amino acids, the NHL repeats at the C-terminus follow the tripartite motif (Figure 3A) [13]. Malin, which is 395 amino acid long, is instead considered a TRIM-like E3 ubiquitin ligase since, unlike canonical TRIM proteins, it lacks the B-box and coiled-coil domains and displays only the RING domain followed by the NHL domain (Figure 3A). TRIM32 and malin are 27% similar overall, and 52% and 38% similar between their RING and NHL domains, respectively (Figure 3A) [29]. From an evolutionary perspective, malin and TRIM32 are related and evolved from a common TRIM ancestor related to TRIM2, TRIM3 and TRIM71 [29] (Figure 3B). Although no crystal structure has been solved for the malin RING domain, structural analysis of the closely related (similarity 52%) TRIM32 RING highlighted the typical globular structure folded around two Zn ions coordinated by cysteine and histidine residues conserved in both proteins (PDB 5FEY) [29,30]. Regarding the C-terminal NHL repeats, no structural data is available for either protein, but the high sequence similarity between TRIM32 and malin would suggest that the NHL domain presents a similar three-dimensional structure. The crystal structure of the NHL domain of the TRIM32 *Drosophila* orthologue, *thin*, was recently solved and shown to fold in the typical beta-barrel shape (PDB 6D69; 6XG7) [31,32]. Using the Swiss-Model homology modeling software (https://swissmodel.expasy.org/, accessed on 1 January 2021) and the crystal structure of the C-terminal domain of *Danio rerio* TRIM71, another NHL-containing TRIM (PDB 6FPT), we modeled the C-terminal domain of both malin and TRIM32 (Figure 3C). According to this model, the NHL domains of both proteins fold in a similar β-propeller structure. Importantly, residues involved in pathological mutations (D487 in TRIM32 and D233 in malin) are located in an analogous position, suggesting a similar potential mechanism of pathogenicity (see below) (Figure 3C). The NHL domain is regarded as a protein interaction module mediating association with substrates.

In the case of malin, the NHL domain is also fundamental for its association with laforin. Different studies show an impairment of malin–laforin complex formation when missense mutations occur in the NHL repeats [33,34]. Given the NHL domains’ similarity in TRIM32 and malin, we could expect a certain level of functional overlap. Indeed, TRIM32 was shown to be able to ubiquitinate some malin substrates in vitro. However, in the same experimental settings, malin displayed no activity towards the tested TRIM32 substrates. Nevertheless, recent studies have shown that both malin and TRIM32 can ubiquitinate the autophagy receptor p62 (see below), thus indicating partially shared physiological functions for the two proteins [29].

### 2.2. Biochemical Activity

TRIM32 ubiquitin ligase activity has been extensively studied both in vitro and in vivo. Recombinant TRIM32 can promote its autoubiquitination in the presence of E2 enzymes of the D subfamily (UbE2D, UbcH5), can form poly-ubiquitin chains with UbE2N (Ubc13)/UbE2V2 (Mms2), and can self-ubiquitinate with a single ubiquitin molecule with the priming E2s UbE2W (Ubc16), UbE2E1 (UbcH6), and UbE2E3 (UbcH9) [35]. This marked E2 specificity is also maintained with respect to the ubiquitination of substrates in vitro, as TRIM32 was indeed shown to ubiquitinate muscular actin in the presence of UbE2D1 (UbcH5a), UbE2D3 (UbcH5c), and UbE2E1 (UbcH6) [36]. The ubiquitin ligase activity of TRIM32 has also been observed in vivo towards a large number of substrates. As inferred by the ability of TRIM32 to cooperate with various E2s in vitro, the outcomes of TRIM32-mediated ubiquitination on physiological substrates are variable and range from monoubiquitination (e.g., p62) to degradative K48-linked ubiquitination (e.g., PB1), or the regulation of signaling by K63-linked ubiquitination (e.g., STING) [37,38,39]. Cumulatively, these results indicate that TRIM32 can cooperate with various E2-conjugating enzymes to target a large number of substrates with great specificity, thus explaining its important role in numerous physiological processes.

In turn, in vitro ubiquitination assays have demonstrated that malin can interact with three different types of E2-conjugating enzymes: UBE2H (UbcH2), UBE2D (UbcH5), and UBE2E1 (UbcH6) [33,40]. Consistent with the observation that malin E3 ubiquitin ligase mediates mainly the formation of K63-linked poly-ubiquitin chains on the target substrates, a subsequent study demonstrated that the E2 UBE2N (Ubc13), a peculiar promoter of K63 poly-ubiquitin linkages [41], associates with malin in vivo [42].

### 2.3. Mutations Affecting TRIM32 and Malin Activity

Several biochemical studies have helped to define the contribution of TRIM32 single domains to its activity. As observed for other TRIM family members, TRIM32 ubiquitin E3 ligase activity relies on the presence of a functional RING domain, which is necessary to mediate interaction with the ubiquitin-loaded E2-conjugating enzyme [36,43]. Correct folding of the RING domain is also a necessary requisite for activity, and indeed, mutations in Zn-coordinating residues, such as the H42A mutation in TRIM32, are sufficient to abolish it [29]. Furthermore, structural studies of isolated TRIM RING domains alone or in complex with ubiquitin-loaded E2s have demonstrated that RING dimerization is necessary for catalysis. Indeed, in dimeric RING-type E3 ligases, the two subunits of RING dimers make contact with both E2 and E2-bound ubiquitin, promoting the stabilization of a closed conformation of the E2-ubiquitin complex, which is then catalytically primed for ubiquitin transfer on the substrate [44,45,46]. Although no structural studies showed the complex between TRIM32 RING and ubiquitin-bound E2, mutational analysis demonstrated that TRIM32 RING also relies on dimerization for activity. Indeed, TRIM32 RING mutant I85R is unable to dimerize and cannot promote the discharge of ubiquitin from E2-conjugating enzymes [30]. Whether dimerization of malin RING is likewise necessary for its activity is currently unknown. Furthermore, besides the RING domain, the B-box and coiled-coil of TRIM32 also play a role in activity modulation. Indeed, the fully active form of TRIM32 is represented by a tetramer formed upon the interaction of the coiled-coil domains, while the B-box domain was shown to modulate the rate of poly-ubiquitin chain synthesis [30,35]. As these domains are absent in malin, it would be interesting to investigate whether this E3 ligase can modulate its activity and the molecular mechanisms involved.

Additional indications on residues/domains relevant for the activity of the two proteins can be gathered from the many pathological mutations occurring in *EPM2B* (malin) and in *TRIM32*, graphically summarized in Figure 4. Most of the LGMDR8-causing mutations are clustered within the C-terminal NHL domain, whereas the LD-causing mutations in *EPM2B* are more homogeneously distributed along the malin protein (Figure 4).

LGMDR8 was initially described in the Hutterite population of North America with NHL domain mutation c.G1459A (p.D487N), identified as a founder mutation in this population [15,47]. Later studies identified many additional mutations in non-Hutterite patients [19,20]. In the case of malin, about 90 different mutations have been identified along *EPM2B* and associated with LD [48] (Figure 4). A database is available, collecting all genetic modifications: http://projects.tcag.ca/lafora, accessed on 1 January 2021. As for TRIM32 NHL mutations likely affecting interaction with its substrates, malin D146N mutation breaks the interaction with laforin without affecting its ubiquitinating activity. In the presence of this mutation, the formation of a laforin–malin functional complex is lost and, as a consequence, there is an aberrant accumulation of glycogen [40,49]. The most common malin mutation is, however, P69A, located in the RING domain and likely compromising its E3 catalytic activity, while to date only the frameshift mutation p.C39LfsX17 has been reported in the RING domain of TRIM32. Additional *TRIM32* mutations in domains other than the RING and NHL were identified in the coiled-coil region in LGMDR8 patients [18,50]. Interestingly, the mutation p.P130S in TRIM32 B-box is associated with Bardet–Biedl syndrome type 11 (BBS11), a multisystemic disorder characterized by retinal dystrophy, obesity, kidney abnormalities, and polydactyly, but with no skeletal muscle involvement [51]. The presence of different diseases associated with *TRIM32* further confirms its pleiotropic role and possible tissue- and cell-type-specific functions, however the pathogenic mechanism linking the TRIM32 B-box mutation with BBS has not been investigated yet. Interestingly, some LGMDR8 patients carrying the premature truncating mutation p.C39LfsX17 also present systemic defects typical of BBS [18]. These observations suggest that mutations in catalytic or regulatory domains (RING and B-box, respectively) cause BBS11, while mutations in the NHL disrupt muscle-specific functions, and the presence of LGMDR8 patients carrying deletions of the whole *TRIM32* coding sequence without BBS symptoms argues against this hypothesis. Furthermore, several reports indicate that mutations in TRIM32 NHL domain lead to remarkable instability in the TRIM32 protein, as observed in patients with the D487N and other NHL mutations and in *Trim32* knock-in mice carrying the corresponding D489N variant [18,52,53]. In this scenario, LGMDR8 is likely due to the total loss of expression of TRIM32 and the effects of TRIM32’s absence may therefore affect also proteins not directly targeted by its E3 ligase activity. However, challenging the hypothesis that *TRIM32* point mutations result in loss of expression, Locke et al. showed that the stability of TRIM32 WT and D487N in transfected COS-7 cells is relatively similar [54]. Nevertheless, even if the exact pathological mechanisms have not been elucidated for either disease, it is likely that most of the pathological mutations identified so far presumably affect or totally abolish TRIM32 and malin function and/or interaction with their substrates.

## 3. TRIM32 and Malin in Normal Physiological Processes

### 3.1. TRIM32

TRIM32 plays a versatile role in cellular functions by affecting a wide range of signaling pathways due to its broad expression and interaction with many cellular substrates. Indeed, TRIM32 mRNA is detected in almost all tissues, including brain and skeletal muscle where it is also expressed during embryonic development, in keeping with its role as a cell fate determinant [15,55,56]. TRIM32 exerts its physiological functions through association with many substrates and interactors, which are summarized in Appendix A and reviewed in Lazzari et al., 2016 [57]. In this review, we will mainly refer to substrates and interacting proteins relevant to LGMDR8 pathogenesis.

#### 3.1.1. Maintenance of Sarcomeric Structures

The ubiquitin proteasome system (UPS) plays critical roles in the maintenance of sarcomeric structures and in the degradation of muscular proteins upon atrophic signals. Studies of the potential role of Trim32 in these pathways have initially shown that it cosediments together with actin, α-actinin, tropomyosin 1 α chain, desmin, and myosin regulatory light chain in the soluble fraction of mouse tibialis anterior (TA) [58]. Furthermore, in the thin filaments of adult muscle cells, Trim32 interacts with myosin in a coiled-coil-dependent manner, although there is no evidence that the interaction results in myosin ubiquitination [36]. Actin can instead be mono- or diubiquitinated by TRIM32 in vitro [36] and indeed, actin can also be ubiquitinated by Trim32 within the intact myofibril isolated from mouse TA [58]. Interestingly, the overexpression of TRIM32 in HEK293 cells results in reduced levels of actin, thus suggesting that actin degradation may be mediated by TRIM32 in various cell types [36]. Of relevance for muscular physiology, Trim32 was also shown to ubiquitinate desmin both in vitro and in vivo. Atrophic signals were shown to promote the phosphorylation of desmin, which is then targeted by Trim32 for ubiquitination and degradation, resulting in the loss of desmin filaments and sarcomere destabilization [58] (Figure 5).

#### 3.1.2. Differentiation and Homeostasis of Satellite Cells

TRIM32 was shown to play an important role in regulating muscle stem cells’ (satellite cells) differentiation and also in maintaining their homeostasis. In murine C2C12 myoblasts, Trim32 was shown to bind to and ubiquitinate c-Myc, promoting its degradation, thus triggering cell-cycle exit and the initiation of the differentiation process. Interestingly, this effect was lost when overexpressing a RING-inactive TRIM32 mutant (C24A) [55,56,59]. Furthermore, the treatment of cells with proteasome inhibitors can prevent the degradation of c-Myc induced by co-expression of TRIM32, suggesting that polyubiquitination of c-Myc promotes its proteasomal-mediated degradation [55]. However, the differentiation of myoblasts does not appear to be significantly affected upon knockdown of c-Myc, suggesting that the c-Myc-mediated pathway is not the only one involved in myoblast differentiation [55]. Indeed, Trim32 was also shown to promote muscle satellite-cell differentiation by regulating the stability of NDRG2, a protein whose silencing in myoblasts is sufficient to induce differentiation. NDRG2 can be polyubiquitinated by TRIM32 in vitro, and, in muscles from *Trim32* KO mice, reduced ubiquitination of NDRG2 was detected, indicating that TRIM32 can control NDRG2 levels through the ubiquitin proteasome system (UPS). The loss of Trim32-mediated degradation of NDRG2 in *Trim32*-deficient myoblasts results in its accumulation, impairing cell-cycle exit and the onset of differentiation [60] (Figure 5).

In addition to the regulation of satellite-cell differentiation, TRIM32 may play a broader role in controlling other aspects of the biology of muscular stem cells. The E3 SUMO ligase Piasy was shown to be ubiquitinated by TRIM32 and targeted for degradation. As elevated SUMOylation levels are a hallmark of cell senescence, TRIM32 activity towards Piasy might be a mechanism to inhibit the senescence of satellite cells and, therefore, to regulate their overall homeostasis [61].

#### 3.1.3. Differentiation and Homeostasis of Motor Neurons

As mentioned previously, the LGMDR8 myopathic phenotype was shown to present a neurogenic component. This prompted studies to investigate the potential roles of TRIM32 in neurons. Interestingly, in neural stem cells (NSCs), TRIM32 was shown to distribute asymmetrically upon cell division and to promote differentiation of the cell inheriting it. From a molecular point of view, the mechanism was shown to be similar to what is observed for satellite cells, i.e., it is driven by the Trim32-mediated degradation of c-Myc leading to cell-cycle exit and differentiation. Indeed, in proliferating NSCs, Trim32 is sequestered in the cytoplasm through PKCξ binding. Upon differentiation stimuli, this interaction is abrogated and Trim32 translocates into the nucleus to initiate differentiation by targeting c-Myc for proteasomal degradation [59]. Furthermore, in neural stem cells, a novel role for Trim32 in the regulation of miRNAs was also demonstrated: through interaction with Argonaute-1, Trim32 can regulate microRNA activity in NSCs, including activating the antiproliferative Let-7a, ultimately promoting differentiation [56] (Figure 5).

### 3.2. Malin

Malin is expressed in all tissues, but especially in the liver, pancreas, heart, skeletal muscle, and brain (mainly in the cortex) [24]. Malin tags ubiquitin substrates meant to be recognized by laforin, a dual-specificity phosphatase, in the laforin–malin complex. To date, several malin substrates have been found and they have been gathered in Appendix A. Of those listed, only the substrates that support the involvement of malin in processes related to LD, in particular glycogen metabolism, will be highlighted.

#### Glycogen Metabolism

During glycogen metabolism, the synthesis and degradation of glycogen are mediated by two fundamental enzymes, glycogen synthetase (GS) and glycogen phosphorylase (PYG). In particular, laforin and malin work together to negatively regulate glycogen synthesis and to improve glycogenolysis [62] (Figure 6A).

Glycogen synthase is the enzyme responsible for the synthesis of glycogen. It transfers glucose monomers from UDP-glucose to the terminal branch of the growing glycogen chain via the formation of α (1→4) glycosidic bonds. The activity of GS is controlled by glycogen synthase kinase-3 (GSK-3), among other protein kinases [63], while its levels are controlled by the laforin–malin complex via ubiquitination. In particular GS is degraded when ubiquitinated by malin. Thus, in LD individuals lacking a functional laforin–malin complex, increased GS levels are observed [64].

In addition to the direct targeting of GS for degradation, malin/laforin can also indirectly regulate their activity. In physiological conditions, the modulation of glycogen accumulation occurs when PP1 (protein phosphatase type 1) is targeted to glycogen by the PTG (protein targeting to glycogen) regulatory subunit, also known as R5, to dephosphorylate and activate GS. PTG/R5 belongs to a group of PP1 regulatory subunits (G_M_, G_L_, PTG/R5, R6, and R3E) with a glycogen-targeting role [65]. Malin–laforin was shown to target PTG/R5 for degradation resulting in GS inhibition. In addition to PTG/R5, R6 and G_L_ are also recognized as substrates of malin. However, the laforin–malin complex is able to inhibit R6-stimulated glycogen accumulation but does not affect the function of G_L_ in this pathway. Taken together, the ubiquitination of PP1 regulatory subunits by malin is necessary to ultimately inhibit glycogen accumulation [66].

Acting as a glucose reservoir, glycogen is stored by the organism in order to be broken down when necessary to be used as a source of energy. The debranching enzyme amyl-1,6-glucosidase,4-α-glucan transferase (AGL) facilitates the breakdown of glycogen by removing the branch points of the sugar to ease the action of glycogen phosphorylase (PYG), an enzyme necessary for glycogenolysis. Although the mechanism of AGL regulation is not clear yet, it appears to be a substrate of malin. When glycogenolysis is increased, AGL is released from glycogen and translocates to the nucleus to be ubiquitinated by malin. In the absence of a functional malin–laforin complex, AGL is not degraded and continues trimming the glycogen molecule, resulting in the accumulation of a linear, and therefore insoluble, glycogen molecule [66].

Finally, AMPK is a serine/threonine-protein kinase that acts as a sensor of cellular energy status and can regulate the activity of the laforin–malin complex. Although the alpha and beta subunits of AMPK are found to be substrates of malin, the K63-linked polyubiquitination of these subunits has no apparent effect on AMPK activity [67]. More investigation is needed to determine the regulation of AMPK by the laforin–malin complex.

## 4. Malin- and Trim32-Related Pathophysiological Mechanisms

The processes described here involving malin and TRIM32 hint at the possible pathological consequences of mutations altering either protein’s function. Indeed, germline mutations in *EPM2B* (malin) and *TRIM32* cause two rare genetic diseases, Lafora disease and Limb-Girdle Muscular Dystrophy R8, respectively. Complete deletions, early frameshift mutations, and the distribution of mutations along the entire length of the protein, whether affecting catalytic activity or the interaction with relevant substrates/partners, likely result in a loss-of-function pathological mechanism compatible with a recessive inheritance of LD and LGMDR8. These are two distinct conditions presenting with characteristic clinical features implicating the two genes in distinct biological processes as described above. Nevertheless, the two proteins share the E3 ubiquitin ligase activity, similar domain composition, and evolutionary origin. Here below, we review pathological mechanisms distinctive of the two diseases and also propose possible common processes.

### 4.1. TRIM32

#### 4.1.1. Muscular Atrophy

As LGMDR8 is a muscle-wasting disorder, initial studies about the pathogenic role of TRIM32 in LGMDR8 were focused on its potential role in muscle atrophy, also given its previously described role in the ubiquitination of muscular sarcomeric proteins [36,58]. However, studies employing different atrophy models (fasting, hindlimb suspension, or denervation) provided controversial evidence regarding the role of TRIM32 in atrophy. For instance, in fasting-induced atrophy some studies revealed that the inhibition of TRIM32 can prevent the loss of myofibrillar proteins, while in another study no differences were observed in body-weight reduction between *Trim32* WT and KO mice, indicating that TRIM32 may not be necessary during muscle atrophy [58,68]. On the other hand, in a mouse hind-limb suspension model, TRIM32 expression was mildly increased during the suspension phase. However, reloading strongly induced significant upregulation of TRIM32, which indicates that TRIM32 is likely involved in muscle regeneration rather than muscle atrophy [36]. Further complicating matters, the potential effects of LGMDR8 mutations in these processes remain to be fully elucidated.

#### 4.1.2. Alteration of Satellite Cells

Lack of regenerating myofibers is one of the hallmarks of LGMDR8. As TRIM32 is involved in the maintenance and differentiation of satellite cells, it is not surprising that its alteration may result in reduced regeneration potential. Indeed, in muscles from *Trim32* KO mice, satellite cells were mislocalized, indicating alterations of their normal physiology, although no clear role for TRIM32 in this process was clarified [55]. Furthermore, TRIM32 loss of function results in accumulation, among other substrates, of the E3 SUMO ligase Piasy and SUMOylated proteins as observed in *Trim32* KO myoblasts. Together with other characteristic metabolic changes, such as the accumulation of senescence-associated β-galactosidase, this evidence points towards a role for TRIM32 in the maintenance of the normal satellite-cell pool necessary to sustain the lifelong regeneration processes characteristic of the healthy neuromuscular system. Besides, the fact that human Piasy fails to colocalize with TRIM32 in fibroblasts isolated from an LGMDR8 patient also supports this study [61,68].

#### 4.1.3. Alteration of Motor Neurons

With regards to the potential role of TRIM32 in neuronal differentiation and homeostasis, this is of particular relevance for pathogenesis. Indeed, denervation induces atrophy and, thus, defects in motor neurons may be reflected in alterations of muscular tissue. Indeed, motor neurons from *Trim32* KO mice display a reduction of neurofilaments, the neuronal-specific intermediate filaments, and also a reduction in axonal diameter. Such axonal alteration may be causative of the observed change in motor unit phenotype towards a slower twitch type, characterized by decreased motor axon caliber and a combined shift from type II (fast) to type I (slow) myosin isoforms in skeletal muscle [17]. However, the molecular mechanism underlying such changes in motor neurons from *Trim32* KO mice has not been elucidated yet.

### 4.2. Malin

#### 4.2.1. Alteration of Glycogen Metabolism

The association of malin with laforin, regulated by AMPK, allows the former to ubiquitinate the latter together with the previously mentioned substrates PTG/R5 and GS. This ubiquitination triggers the release of all three enzymes (laforin, PTG, and GS) from the glycogen particle, directing them towards proteasome-dependent degradation, while the released glycogen molecule moves towards its normal metabolism (Figure 6A). In Lafora disease, glycogen metabolism is impaired, leading to the generation of an abnormal form of glycogen that cannot be degraded. As the disease progresses, this aberrant glycogen builds up, forming aggregates of polyglucosan called Lafora bodies [69]. These aggregates are found at the level of muscles, heart, liver, and brain and have damaging effects especially within the nervous system [70]. As mentioned previously, the activation of glycogen synthesis is regulated by GS dephosphorylation mediated by the PP1 phosphatase, which is recruited to glycogen through the action of one out of five different regulatory subunits (G_M_, G_L_, PTG/R5, R6, and R3E). Previous studies have focused on investigating whether malin and laforin specifically inhibit PTG/R5-stimulated glycogen accumulation or if they inhibit the function of other PP1 regulatory subunits [40]. PTG/R5 (encoded by the *PPP1R3C* gene) and R6 (encoded by the *PPP1R3D* gene) are expressed in different tissues, including the brain. The other three PP1 targeting subunits are expressed in skeletal muscle, and/or heart and liver tissue. Among these subunits, G_L_ (encoded by the *PPP1R3B* gene) is expressed in muscle and liver tissue [65]. Malin and laforin inhibit R6-stimulated glycogen accumulation, but they do not inhibit the function of G_L_. Therefore, malin and laforin inhibit PP1 regulatory subunit-stimulated glycogen accumulation by acting on some PP1 regulatory subunits but not others [66].

In addition to the glycosyltransferase activity required for building the glycogen molecule, GS also incorporates a phosphate (P) in the glycogen on around 1/10,000 glucose monomers. Laforin binds to glycogen via its CBM (carbohydrate-binding module) domain and mediates the removal of the phosphate from the sugar [71]. Hyperphosphorylated glycogen alters the normal function of enzymes responsible for the branching and debranching of glycogen, generating not only long chains but also aberrant forms of sugar, which will prove to be insoluble and, when aggregating, will form the so-called Lafora bodies [72]. In this context, the function of laforin appears to be fundamental to the maintenance of a low level of glycogen phosphorylation and to avoid sugar precipitation. Malin, whose ubiquitinating action is dependent on laforin, seems to regulate the homeostasis of different enzymes involved in glycogen synthesis and of laforin itself, promoting their ubiquitination with consequent degradation at the right moment. In this scenario, a lack of malin functionality would cause an increase in the levels of PTG, GS, and laforin [73]. As a consequence, the formation of Lafora bodies is determined and the enzymes themselves would be sequestered by the inclusion bodies (Figure 6A).

#### 4.2.2. Clearance of Protein Aggregates

In-depth studies on the pathology of LD aimed at investigating the alteration of processes not related to glycogen metabolism in order to identify other potential roles for malin. Several studies support the hypothesis that malin and laforin, by forming a complex with Hsp70, may participate in protein clearance, thus controlling cytotoxic conditions generated by the accumulation of misfolded proteins [74] (Figure 6B). In detail, laforin acts as a bridge between Hsp70 and misfolded proteins to recruit malin, which, in turn, favors the ubiquitination of these proteins, leading to their degradation through the proteasome system [25,73,74]. Studies show that, as a consequence of malin absence, in LD, there is an increase in toxicity caused by misfolded proteins. Besides, the impairment of autophagy due to malin absence may worsen the condition by further inhibiting protein clearance mediated by the laforin–malin complex (see below).

#### 4.2.3. Heat Shock Response

Laforin and malin form a complex with CHIP, a U-box E3 ligase, and a cochaperone that translocates to the nucleus during heat shock conditions to regulate the activation of heat shock factor 1 (HSF-1). By binding to and activating the heat-shock elements present in the promoter region of genes encoding heat shock proteins, HSF-1 protects the cell from heat-shock-induced death conditions [75]. Upon thermal stress, the laforin–malin complex translocates into the nucleus, requiring both CHIP and HSF1 for translocation [75]. Once inside the nucleus, the laforin–malin complex acts as a regulator of HSF1 transcriptional activity (Figure 6C). The mechanism of this transcriptional activation needs further investigation, but studies conducted so far suggest that some pathological symptoms of LD may originate in the HSF-1-mediated stress response pathway defect [75]. Indeed, in the absence of malin, the nuclear translocation of laforin and CHIP is inhibited, and the heat shock response is thusly abolished [76].

#### 4.2.4. Neuroinflammation

The latest studies have shown that elevated levels of proinflammatory mediators are present in the astrocytes and microglia of mouse animal models for LD [77,78], suggesting progressive neurodegeneration, also supported by the reduced viability of neurons [77]. The rationale for increased inflammation in the Lafora disease condition may be associated with the manifestation of epilepsy, as many studies have shown the existence of a relationship between epilepsy and inflammation [79]. However, despite the high levels of proinflammatory mediators such as TNFα, IL-6 and IL-1β, chemokines, cytokines, and other inflammatory markers [77], it is still unknown what exactly drives neuroinflammation in LD (Figure 6D). Some studies indicate that the Lafora bodies accumulating in astrocytes and in the microglia could trigger a stressful condition by promoting the release of proinflammatory mediators, chemokines, and cytokines, and that the increased inflammation causes damage to the neurons. Furthermore, the worsening of inflammatory conditions with age may further exacerbate the disease over time, while treatment with anti-inflammatory agents may represent a means to delay disease progression [77,78,80,81].

## 5. Malin and TRIM32 Common Pathways?

The evolution of paralogue genes is often associated with the conservation of molecular functions that have redefined their specificity. In the case of malin and TRIM32, one obvious conserved biochemical function is the RING-mediated ubiquitin E3 ligase activity, but we can speculate that the shared NHL domain may mediate the involvement of malin and TRIM32 in similar/common processes.

### 5.1. Autophagy Regulation

One of the pathways regulated by both malin and TRIM32 is autophagy, which indeed involves several TRIM family members [82]. Evidence reports that both malin and TRIM32 are positive regulators through the ubiquitination of different components of the autophagy pathway, including phosphatidylinositol 3-kinase catalytic subunit type 3 (PI3KC3) complex and p62.

Different forms of the PI3KC3 complex play a role in multiple membrane trafficking pathways. In particular, PI3KC3-C1 is involved in the initiation of autophagosome formation, and PI3KC3-C2 in the maturation of autophagosomes and endocytosis. The laforin–malin complex physically interacts with core components of the PI3KC3 system and specifically polyubiquitinates Beclin1, Vps34, and Vps15 with K63-linked ubiquitin chains, thus promoting the initial steps of autophagosome formation (Figure 7A). In addition, malin promotes the polyubiquitination of other PI3KC3 components, such as ATG14L and UVRAG. [83]. In Lafora disease, the absence of a functional laforin–malin complex results in reduced activation of the PI3KC3 complex, leading to the observed impairment of the initial steps of autophagy [83]. In keeping with this, in animal models KO for either malin or laforin, and in cell lines derived from Lafora patients, the autophagy process is impaired [84,85].

A key component of the autophagy pathway is p62, also known as sequestosome-1, which is encoded by the *SQSTM1* gene. p62 is an autophagosome receptor targeting ubiquitinated proteins and directing them towards selective autophagy [86]. p62 interaction with ubiquitinated cargo proteins is mediated by its C-terminal ubiquitin-binding domain (UBA), while its short LIR (LC3-interacting region) sequence interacts with LC3, a protein found on the membrane of autophagosomes [87,88]. p62 binds to the laforin and malin complex, and is a substrate of the latter (Figure 7B). A positive feedback loop is then established, with malin ubiquitin ligase activity enhanced by p62 binding, resulting in further p62 ubiquitination, which ultimately promotes p62 recruitment and its ability to target substrates towards autophagy [42].

Interestingly, autophagy alteration has also been detected in muscle samples from LGMDR8 patients, as evidenced by a reduction in p62 and LC3II levels. Furthermore, histological analyses of muscle biopsies from some LGMDR8 patients show the presence of membrane-bound vacuoles containing cytoplasmic degradation products [18]. Whether these changes result from the elevated formation of autophagosomes or the stalled fusion of autophagosomes with lysosomes is unknown, and it is thus not clear whether up- or down-regulation of autophagy occurs in LGMDR8. As observed for malin, TRIM32 can regulate autophagy by acting on p62. In C2C12 myoblasts, p62 is monoubiquitinated by Trim32, and the loss of Trim32 impairs p62 sequestration, while the reintroduction of Trim32 facilitates p62 speckle formation and its autophagic degradation [38] (Figure 7B).

In addition, TRIM32 is conveyed to ULK1, a kinase involved in the initial steps of autophagosome formation, by the autophagy cofactor AMBRA1, resulting in the stimulation of AMBRA1 kinase activity through the synthesis of unanchored K63-linked poly-ubiquitin chains. Notably, the TRIM32 D487N mutation disrupts its ability to bind ULK1 and to induce autophagy in myoblasts (Figure 7A) [89]. As some studies indicate that dysfunctional autophagy can abolish cell differentiation and cause muscle atrophy in mouse models, TRIM32 involvement in autophagy regulation awaits further elucidation, in particular in the context of LGMDR8 [90,91,92]. Furthermore, some studies have suggested that, in addition to being an autophagy regulator, TRIM32 itself may also be regulated through this process. This is of particular relevance given the observed reduced levels of mutated TRIM32 protein in some LGMDR8 patients and in *Trim32* D489N mutant knock-in (KI) mice recapitulating the disease. TRIM32 is itself a substrate of p62 and is targeted to selective autophagic degradation in C2C12 cells. When autophagy was inhibited using Baf-A1 in TRIM32^V591M^ myoblasts isolated from one LGMDR8 patient, the mutant TRIM32 protein level was efficiently rescued, thus suggesting that mutated TRIM32 is degraded through this pathway. However, another study revealed that mutant TRIM32^R394H^ and TRIM32^V591M^ expressed in C2C12 cells do not localize at the lysosome, and that TRIM32^D487N^ is not degraded by autophagy in HEK293 cells under either normal or starvation conditions [18,38]. These conflicting results suggest that possibly different degradation pathways exist between human muscle cells and other cell types, or that the degradation of WT and mutant TRIM32 may depend on different pathways. Nevertheless, both malin and TRIM32 seem to affect autophagy at the level of the initiation step and to share the ubiquitination of p62 through their E3 ligase activity. However, for both malin and TRIM32 further investigations must be carried out to further elucidate their role in autophagy regulation.

### 5.2. Regulation of WNT Pathway

The identification of novel malin substrates is a useful tool to shed light on its possible roles. Malin was for instance shown to be involved in the Wnt signaling pathway through the regulation of the degradation of Dishevelled2 (Dvl2) [93]. Dvl2 is a cytosolic protein regulating the translocation of β-catenin to the nucleus which, in turn, mediates the transcription of Wnt target genes [94]. The loss of malin function results in enhanced the Wnt signaling pathway in the brain, which, in LD patients, correlates with impaired synaptic differentiation and plasticity and other neurogenic defects [93].

Likewise, TRIM32 may also regulate satellite-cell differentiation by activating the Wnt/β-catenin pathway or by potentially affecting Wnt/β-catenin downstream miRNAs, such as miR-133b and miR-206, which promote Pax7 downregulation and, hence, adult muscle progenitor cell differentiation during muscle regeneration [95,96,97]. However, how and whether this is related to the occurrence of LGMDR8 is presently unknown.

### 5.3. Regulation of Glucose Metabolism

Of particular relevance for both neural and muscular tissues, where energetic requirements by neuronal processes and muscular contraction require tight control of glucose metabolism, both malin and TRIM32 were shown to be involved in this pathway. The role of malin, in complex with laforin, in glycogen catabolism has been extensively described above. Interestingly, in the *D. melanogaster* model, the TRIM32 orthologue *thin* was recently shown to be implicated in glucose metabolism through the positive regulation of glycolytic enzymes aldolase and phosphoglycerate mutase [31]. In addition, TRIM32 silencing in normal muscles increases PI3K-AKT-FOXO signaling and enhances glucose uptake, thus inducing fiber growth [98]. Moreover, in cell-culture systems expressing TRIM32 or malin, the former was able to ubiquitinate typical malin substrates such as PTG/R5 and AMPK subunits, albeit with a different chain topology [29]. Whether this is physiologically relevant is at present unknown. Of note, glycogen can contribute to the formation of protein–polysaccharide complexes in muscle. For example, sarcoglycans, linking the actin cytoskeleton with the extracellular matrix to anchor the sarcolemma during muscle contraction, are altered by TRIM32 ablation in both the fruitfly and C2C12 cell models [31,32]. Taken together, these findings indicate that common pathways controlled by malin and TRIM32, even in a contrasting manner, may deserve further investigation to potentially identify novel therapeutic targets for both LD and LGMDR8.

## 6. Perspective

Considering phylogenetic analyses, malin and TRIM32 appear to share a common evolutionary origin. In particular, malin seems to have evolved from an ancestral TRIM by acquiring the E3 ubiquitin ligase activity. However, the peculiar structural differences between malin and TRIM32 deeply influence their respective substrate specificity. Indeed, malin is not able to ubiquitinate TRIM32 substrates, but, for some substrates, the opposite occurs. This could depend on the lack of B-box and coiled-coil domains in malin. In addition, for the identified shared substrates, the type of ubiquitination catalyzed by the two proteins can be different, with important consequences in a pathological setting.

Taking into account that the diseases caused by malin and TRIM32 lead to neurological defects, it is plausible to interrogate if, in addition to the already known substrates that determine these conditions, others may be common to both proteins. Pursuing such a goal in the future could open many scenarios aimed at improving not only the specific pathological conditions related to these two E3 ubiquitin ligases but also neurodegeneration and neuromuscular disorders from a broader perspective. Searching for novel substrates of malin and/or TRIM32 could lead to the discovery of elements also involved in pathways not related to the two proteins, but in common with other types of TRIM proteins with E3 ubiquitin ligase function.

Advancements in basic knowledge of malin and TRIM32 activity and disease mechanisms are also necessary to envisage therapeutic strategies. The identification of small molecules that can specifically modulate either the activity or the interaction with substrates/interactors or the specific deubiquitinating enzyme that counteracts the activity of these two ubiquitin ligases could represent promising drug development approaches.

## Figures and Tables

**Figure 1 cells-10-00820-f001:**
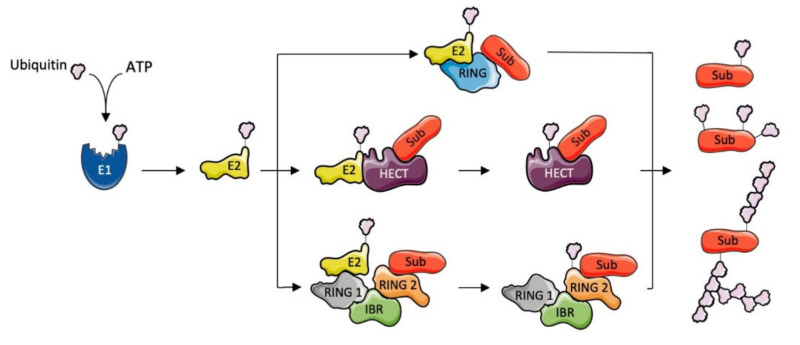
The ubiquitination process and E3 ubiquitin ligases. Protein ubiquitination is carried out by a ubiquitin (Ub)-activating enzyme (E1), a Ub-conjugating enzyme (E2), and a Ub-ligating enzyme (E3). The E1 activates Ub in an ATP-dependent manner through the formation of a thiol-ester bond; the activated Ub is transferred to the Cys-active site of an E2; the E3 facilitates ubiquitination of the substrate and defines substrate specificity. The different mechanisms of the three main classes of E3 Ub ligases is depicted. The HECT (homologous to the E6AP carboxyl terminus) domain-containing E3s form a thiolester with Ub, which is then transferred to the amino group of a substrate lysine. Non-HECT domain E3s containing a catalytic RING or RBR (RING-In between RING-RING) function as adaptors linking substrate and Ub-charged E2. Targeting different Ub lysines or Met1 for chain formation allows the generation of a great variety of distinct Ub modifications, including different homotypic chains, and mixed and branched chains.

**Figure 2 cells-10-00820-f002:**
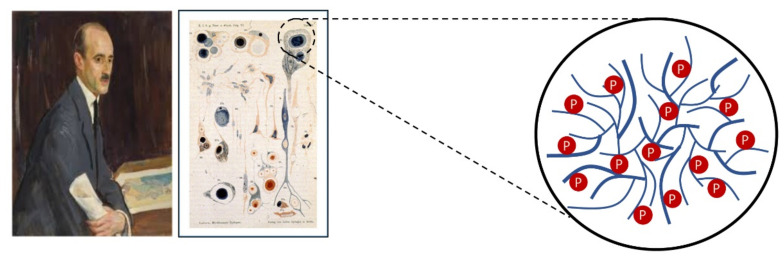
In 1911, the Spanish neurologist Dr. Gonzalo Rodriguez Lafora described polyglucosan bodies in the perikaryon of neurons in the cerebral cortex, thalamus, globus pallidus, and substantia nigra of affected patients. Lafora bodies are the result of the aggregation of long, unbranched, hyperphosphorylated and insoluble forms of glycogen. The left panel shows a painting of Dr. Lafora displayed at Prado Museum (Madrid). The middle panel shows a copy of the original paintings of neurons from affected patients. The right panel shows a cartoon of the hyperphosphorylated aberrant glycogen present in Lafora bodies.

**Figure 3 cells-10-00820-f003:**
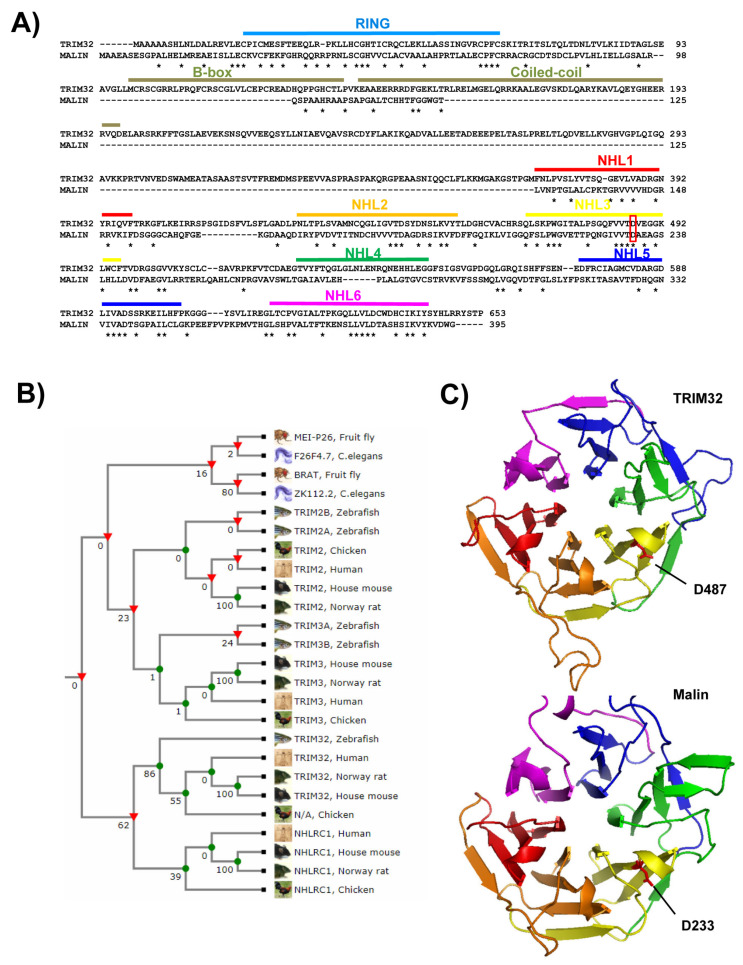
Structural data of TRIM32 and malin. (**A**) Protein sequence similarity between TRIM32 and malin. The position of the RING domain (in blue), the B-box and coiled-coil domains (in grey), and the six NHL domains (in red, orange, yellow, green, dark blue, and magenta) are indicated; the position of the residues D487 and D233 is also indicated in a red box; protein sequence alignment was performed using CLUSTAL-W (https://www.expasy.org/resources/clustalw, accessed on 1 January 2021). Identical amino acids are indicated with an asterisk at the bottom of the sequence. (**B**) Phylogenetic relationship between TRIM32 and malin (NHLRC1) using TreeFam from EMBL-EBI. (**C**) Modelling of the C-terminal domain of TRIM32 and malin using the Swiss-Model homology modeling software (http://swissmodel.expasy.org, accessed on 1 January 2021) and the crystal structure of the C-terminal domain of *Danio rerio* TRIM71 (PDB 6FPT) as a model. Disease-related residues D487 in TRIM32 and D233 in malin are located in an analogous position.

**Figure 4 cells-10-00820-f004:**
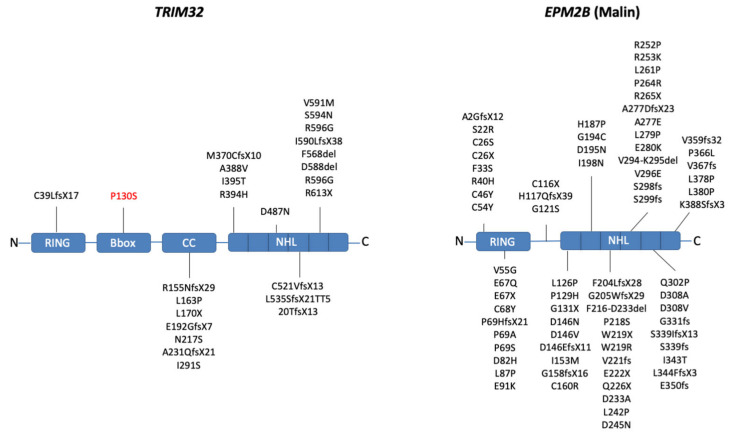
Graphical summary of the pathological mutations occurring in TRIM32 (left) and EPM2B (malin; right) genes. Domain names are indicated and the mutations within the NHL domains are grouped according to single NHL repeats. Most LGMDR8-causing mutations cluster within the C-terminal NHL domain of TRIM32 and may be in common with the allelic disease sarcotubular myopathy (STM), while the P130S mutation in the B-box (red) is associated with BBS11. LD-causing mutations in EPM2B are more homogeneously distributed.

**Figure 5 cells-10-00820-f005:**
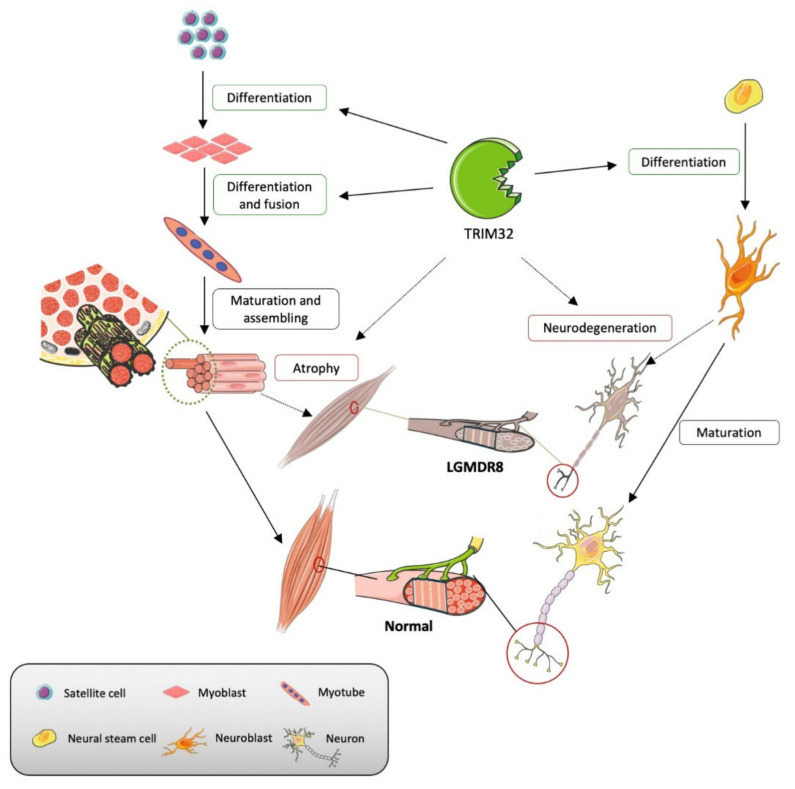
Limb-girdle muscular dystrophy R8 (LGMDR8) and the role of TRIM32 in the context of muscle and neuronal differentiation/degeneration. The activated satellite cells differentiate into myoblasts and then fuse together to form myotubes. The myotubes further mature and assemble into myofibers. The neural stem cells are activated and differentiate into neuroblasts, then further mature into neurons. TRIM32 is involved in these physiological processes through regulating differentiation and fusion (green boxes). Dysfunction of TRIM32 will disrupt normal neuromuscular system growth and possibly induce atrophy and neurodegeneration, which causes the occurrence of LGMDR8 (red boxes).

**Figure 6 cells-10-00820-f006:**
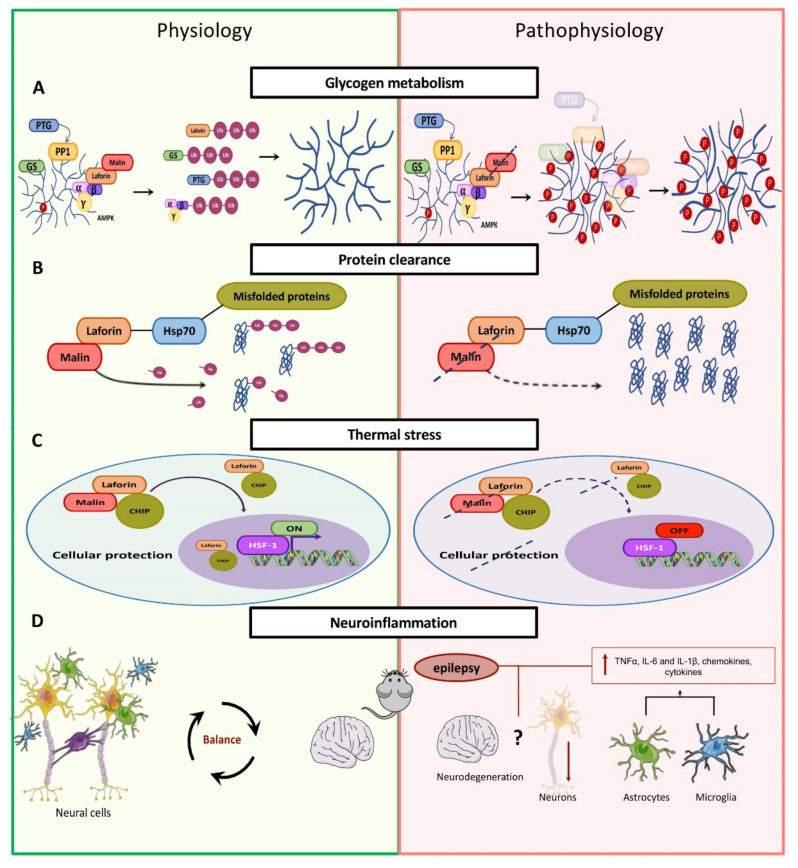
Physiological and pathophysiological processes related to malin. Panel (**A**), physiological negative regulation of glycogen synthesis through the cooperation of laforin and malin. On the right, formation of hyperphosphorylated glycogen as a consequence of the increase in the levels of PTG, GS, and laforin. Panel (**B**), misfolded protein clearance or aggregation depending on the functional synergy between malin and laforin. Panel (**C**), formation of the ternary laforin–malin–CHIP complex to cope with thermal stress through the transcriptional activation of HSF1. Panel (**D**), mouse animal models for LD show the reduced viability of neurons and elevated levels of pro-inflammatory mediators triggered by astrocytes and microglia.

**Figure 7 cells-10-00820-f007:**
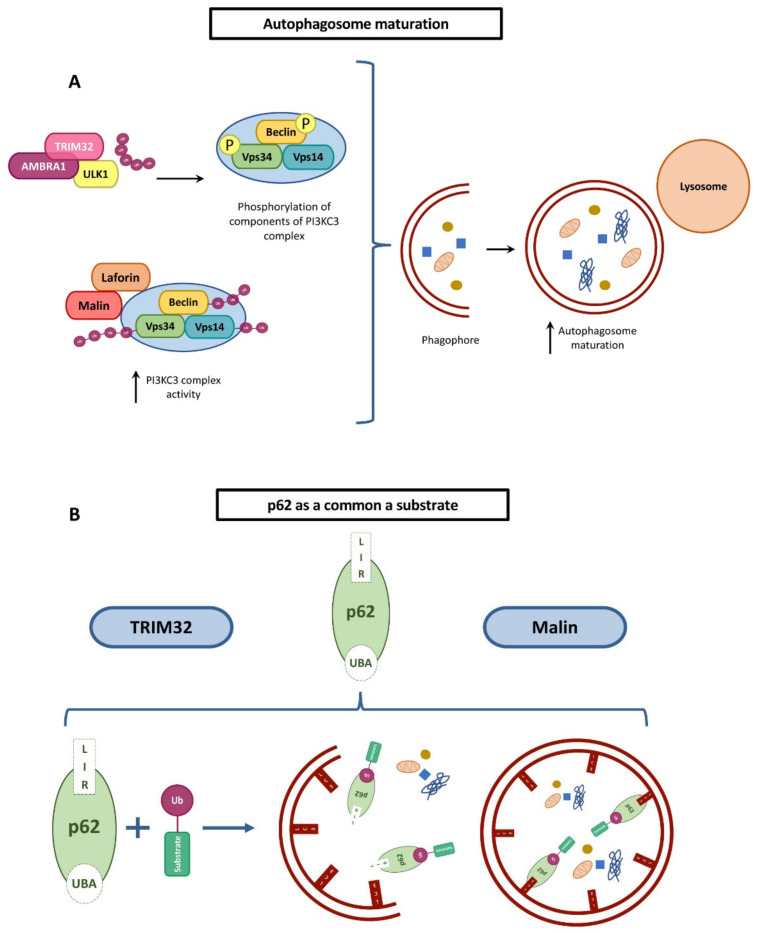
Common roles of TRIM32 and malin in regulating autophagy. Panel (**A**), initial steps of autophagosome formation. The laforin–malin complex regulates the initial steps of autophagosome formation through the assembly of a stable PI3KC3 complex. Similarly, TRIM32 affects the initial steps of autophagosome formation by regulating ULK1 function. Panel (**B**), p62 is a common substrate of TRIM32 and malin. The p62 autophagy receptor is involved in the recognition of ubiquitinated cargo proteins through the UBA domain. Then, p62 recruits the cargo proteins to autophagosomes via its LIR domain, which interacts with LC3. p62 is ubiquitinated by TRIM32 and also by malin, but this modification leads to different outcomes, degradation of p62 in the case of TRIM32 and improvement of autophagy in the case of malin.

## Data Availability

Not applicable.

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
