# Peer review of "TRIM32 and Malin in Neurological and Neuromuscular Rare Diseases"

_cells, 2021, doi:10.3390/cells10040820_

Round 1

Reviewer 1 Report

The review " TRIM32 and malin in neurological and neuromuscular rare diseases” is a review about the function of both E3 ubiquitin ligases. In general the review is interesting and well organized but I have some minor concerns:

Minor points:

  • The tittle of section 2 is Structural data and biochemical function. Authors should specified more, for example: Structural data and biochemical function of TRIM3 and malin
  • A table about diseases and disorders originated by TRIM32 or malin may be illustrative of its importance.

Author Response

The review " TRIM32 and malin in neurological and neuromuscular rare diseases” is a review about the function of both E3 ubiquitin ligases. In general the review is interesting and well organized but I have some minor concerns:

Minor points:

The tittle of section 2 is Structural data and biochemical function. Authors should specified more, for example: Structural data and biochemical function of TRIM3 and malin

We have change the title of section 2 as indicated by the reviewer (page 5).

A table about diseases and disorders originated by TRIM32 or malin may be illustrative of its importance.

In the case of malin, only one disease is related to defects in this protein, namely progressive myoclonus epilepsy type 2 (Lafora disease). For what concerns TRIM32, as we already have 7 figures and 2 large supplementary tables, we choose to highlight the only mutation in TRIM32 (P130S) causing Bardet-Biedl type 11 (BBS11) in Figure 4 and in the relative caption. In the legend of the same figure, we also stressed that LGMDR8 and STM are allelic and the same mutation can result in the two different clinical conditions.

Reviewer 2 Report

This review article has significance and important to know TRIM proteins. To improve this article, I suggest the following points.

1: The expression pattern in development and organs about TRIM32 and Malin would be important to understand their function and pathological importance.

2: In Figure 3, similarity or identity score should be shown.

3: In Figure 4, "NHL domain" should be represented in the figure. The authors should mark both TRIM32 and Malin clearly not to confuse the readers.

4: The proteins which interact with TRIM32 and Malin should be written in Figure 4 or an additional figure to represent which protein bind to which domain in TRIM32 or Malin. This figure would help the readers understand this review.

Author Response

This review article has significance and important to know TRIM proteins. To improve this article, I suggest the following points.

1: The expression pattern in development and organs about TRIM32 and Malin would be important to understand their function and pathological importance.

We have included the expression pattern of TRIM32 (page 9) and malin (page 12) in the revised manuscript. Now it says: “TRIM32 mRNA is detected in almost all tissues, including brain and skeletal muscle where it is also expressed during embryonic development in keeping with its role as a cell fate determinant [15], [55], [56]” (page 9), and “Malin is expressed in all tissues, but especially in liver, pancreas, heart, skeletal muscle and brain (mainly in cortex) [24](page 12).

2: In Figure 3, similarity or identity score should be shown.

In page 5 we have included the sentence “TRIM32 and malin are 27% similar overall, and 52% and 38% similar between their RING and NHL domains respectively (Figure 3A) [29]”. In addition in legend of Figure 3, we have included the sentence “protein sequence alignment was performed using CLUSTAL-W (https://www.expasy.org/resources/clustalw). Identical amino acids are indicated with an asterisk at the bottom of the sequence”.

3: In Figure 4, "NHL domain" should be represented in the figure. The authors should mark both TRIM32 and Malin clearly not to confuse the readers.

We added the headings (TRIM32 and EPM2B) above the schemes in Figure 4 and substituted the 6 NHL repeats with a box representing the full domain (Page 8). The legend of Figure 4 has been changed accordingly.

4: The proteins which interact with TRIM32 and Malin should be written in Figure 4 or an additional figure to represent which protein bind to which domain in TRIM32 or Malin. This figure would help the readers understand this review.

We agree with the reviewer that a figure indicating the parts of TRIM32 and malin involved in the interaction with different partners will help to understand the action of these two E3 on them. Unfortunately, this information is very scarce and only in very few partners the domains involved in the interaction are known. When available, for TRIM32 the information is in Supplementary Table 1. For this reason, with all our respects, we would prefer not to include this information in the figure.

Reviewer 3 Report

The authors Kumarasinghe et al in their manuscript entitled ‘TRIM32 and malin in neurological and neuromuscular rare diseases’ have extensively documented the role of TRIM32 and malin E3 ligases in Limb-Girdle Muscular Dystrophy R8 and Lafora disease respectively. The authors chose these two E3 ligases because they are evolutionarily related, descend from a common ancestor, and both display similar structural domains (NHL).

The review is an in-depth description of TRIM32 and malin in terms of their biochemical activity, mutations in diseases, as well as their function in normal physiological conditions. The authors further discuss how the two proteins share common pathogenic pathways.

Overall, the article is exhaustive and discusses the functions of TRIM32 and malin in great detail. The review includes good illustrations at appropriate places which make the respective points very clear.   

Author Response

The authors Kumarasinghe et al in their manuscript entitled ‘TRIM32 and malin in neurological and neuromuscular rare diseases’ have extensively documented the role of TRIM32 and malin E3 ligases in Limb-Girdle Muscular Dystrophy R8 and Lafora disease respectively. The authors chose these two E3 ligases because they are evolutionarily related, descend from a common ancestor, and both display similar structural domains (NHL).

The review is an in-depth description of TRIM32 and malin in terms of their biochemical activity, mutations in diseases, as well as their function in normal physiological conditions. The authors further discuss how the two proteins share common pathogenic pathways.

Overall, the article is exhaustive and discusses the functions of TRIM32 and malin in great detail. The review includes good illustrations at appropriate places which make the respective points very clear.

We thank the reviewer very much for his/her positive comments on our work.